# Novel bioassay to assess antibiotic effects of fungal endophytes on aphids

**Nicholas Paul Collinson** [1,2]*, **Ross Cameron Mann**[1], **Khageswor Giri**[1], **Mallik Malipatil**[1,2], **Jatinder Kaur**[1], **German Spangenberg**[1,2], **Isabel Valenzuela**[1]

**1** Agriculture Victoria Research, AgriBio Centre for AgriBioscience, Bundoora, Victoria, Australia, **2** School of Applied Systems Biology, Department of Science, Health and Engineering, La Trobe University, Bundoora, Victoria, Australia

* nicholas.collinson@agriculture.vic.gov.au

**Data Availability Statement:** All relevant data are within the manuscript.

## Abstract

Perennial ryegrass is an important feed base for the dairy and livestock industries around the world. It is often infected with mutualistic fungal endophytes that confer protection to the plant against biotic and abiotic stresses. Bioassays that test their antibiotic effect on invertebrates are varied and range from excised leaves to whole plants. The aim of this study was to design and validate a "high-throughput" in-planta bioassay using 7-day-old seedlings confined in small cups, allowing for rapid assessments of aphid life history to be made while maintaining high replication and treatment numbers. Antibiosis was evaluated on the foliar and the root aphid species; *Diuraphis noxia* (Mordvilko) and *Aploneura lentisci* (Passerini) feeding on a range of perennial ryegrass–*Epichloë festucae* var. *Lolii* endophyte symbiota. As expected, both *D. noxia* and *A. lentisci* reared on endophyte-infected plants showed negatively affected life history traits by comparison to non-infected plants. Both species exhibited the highest mortality at the nymphal stage with an average total mortality across all endophyte treatments of 91% and 89% for *D. noxia* and *A. lentisci* respectively. Fecundity decreased significantly on all endophyte treatments with an average total reduction of 18% and 16% for *D. noxia* and *A. lentisci* respectively by comparison to non-infected plants. Overall, the bioassay proved to be a rapid method of evaluating the insecticidal activity of perennial ryegrass–endophyte symbiota on aphids (nymph mortality could be assessed in as little as 24 and 48 hours for *D. noxia* and *A. lentisci* respectively). This rapid and simple approach can be used to benchmark novel grass–endophyte symbiota on a range of aphid species that feed on leaves of plants, however we would caution that it may not be suitable for the assessment of root-feeding aphids, as this species exhibited relatively high mortality on the control as well.

## Introduction

Perennial ryegrass, *Lolium perenne* (L.), is one of the most important pasture grass species globally [1]. It is favoured over other grass species, particularly in southern Australia, as it is highly nutritious, can grow with minimal soil moisture during dry conditions and can remain

**Funding:** NPC was supported by a research scholarship from DairyBio (dairybio.com.au), a joint venture of the Victorian Department of Jobs, Precincts and Regions (Agriculture Victoria Research), Dairy Australia and The Gardiner Foundation. The funder provided support in the form of salaries for NPC, but did not have any additional role in the study design, data collection and analysis, decision to publish, or preparation of the manuscript. The specific roles of these authors are articulated in the 'author contributions' section.

**Competing interests:** NPC received a salary from DairyBio for the duration of this study, as part of a stipend for a PhD project. This does not alter our adherence to PLOS ONE policies on sharing data and materials.

productive for up to four years [1,2]. Like any domesticated crop, however, perennial ryegrass is prone to a range of biotic and abiotic stresses that reduce its yield and persistence [3].

Nematoda, Mollusca, Collembola, Acari, and Insecta (Coleoptera, Lepidoptera, Orthoptera, Diptera, Thysanoptera and Hemiptera) are among the main invertebrate groups of significance that negatively impact perennial ryegrass crops in Australia and around the world predominantly through feeding injuries and the transmission of plant diseases [4,5]. The damage caused by invertebrate pests is mostly quantified in terms of the loss of dry matter or seed yield the extent of which varies depending on the pest type and abundance, as well as the grass species, its growth stage and the part of the plant that is consumed [6]. The most common metric used to assess crop damage by pests is the resultant economic loss [7]. For instance, in New Zealand, grass grubs, *Costelytra zealandica* (White), have been reported to cause productivity losses of $140–380 million NZD on dairy farms, while Argentine stem weevil, *Listronotus bonariensis* (Kuschel), is estimated to cause annual productivity losses of up to $200 million NZD [7]. Often, farmers attempt to offset this high economic cost through the increased use of insecticides and/or increased sowing frequency. Depending on the severity of the damage, farmers may need to supplement herds' diets with silage or grains, further increasing the costs. As a result, the overall economic impact that invertebrate pests have on dairy pastures in New Zealand has recently been estimated to reach up to $1.4 billion NZD [7]. This study found that the bulk of the losses were caused by scarab beetles (up to $600 million approx.), nematodes (up to $300 million approx.) and weevils ($200 million approx.) [7]. The overall economic loss caused by invertebrates on dairy pastures in Australia is unknown.

One group of invertebrates that causes serious damage to arable and pasture crops are aphids, due to direct feeding injuries and virus-associated injuries [8]. Common aphid species found in pasture grasses in Australia and around the world are *Rhopalosiphum* spp., *Sitobion* spp., *Metopolophium* spp., *Schizaphis graminum* (Rondani), *Diuraphis noxia* (Mordvilko) and *Aploneura lentisci* (Passerini) [9]. All these aphids have the potential to cause serious damage to pasture crops in Australia except for *S. graminum* that is not found in Australia. *Rhopalosiphum* spp., *Sitobion* spp. and *Metopolophium* spp. are vectors of Luteoviruses that can cause significant yield losses in perennial ryegrass [10], while *D. noxia*, a recent arrival to Australia, and *A. lentisci* have the potential to cause significant damage due to direct feeding injuries [11]. Farmers use systemic insecticides or spray to control aphids particularly when the crop is at the seedling stage when it is most at risk of virus infection. Despite the relative success of insecticides in controlling foliar aphids, the control of root-feeding aphids such as *A. lentisci* is more difficult. Farmers are also consciously moving away from using insecticides, due to concerns related to increased resistance in aphid populations, increased mortality of natural enemies, along with general health and environmental concerns [12,13]. Hence, there is global interest in the development of innovative pest control solutions, particularly the use of endophytic fungi (e.g. *Acremonium* spp., *Epichloë* spp.) that occur naturally in grasses and are known to protect them from biotic and abiotic stresses [14].

Fungal endophytes have beneficial effects on the host plant as they play an important role in its survival, protecting it from invertebrates and grazing animals through the production of certain chemical compounds, and providing resistance to environmental extremes such as drought or high temperatures [14,15]. However, the same fungal endophytes can be harmful to grazing livestock as these chemical compounds can cause a range of toxicoses such as ryegrass staggers in cattle and sheep, costing the livestock industry millions of dollars in productivity losses [14,16]. These compounds, that confer resistance to vertebrates and invertebrates, are alkaloids of various classes consisting of: ergopeptines (e.g. ergovaline), indole diterpenes (e.g. lolitrem B and epoxy-janthitrems), pyrrolizidines (e.g. lolines), and polyketides (e.g. peramine) [17–19]. These alkaloids have various degrees of toxicity toward vertebrates and

**Table 1. Aphid species collection details and GenBank accession numbers.**

| Aphid species | GenBank Accession number | Locality | GPS | Date of collection | Host plant |
|---|---|---|---|---|---|
| *Diuraphis noxia* | MN066606 | Horsham, Victoria | 36°43'17.8"S 142°10'26.9"E | 1/06/2016 | Barley |
| *Aploneura lentisci* | MN066607 | Bundoora, Victoria | 37°43'05.2"S 145°02'49.6"E | 10/06/2017 | Perennial ryegrass (cv. Impact 04) |

invertebrates [18–20]. Developing endophyte strains that produce favourable quantities and types of alkaloids that remain harmless to grazing animals but retain the toxicity towards invertebrates has been the focus of recent research, particularly in Australia with *Epichloë festucae* var. *lolii* in perennial ryegrass [21–23].

Numerous studies have been carried out since the 1980s to determine what grass–endophyte association has the greatest negative effect on aphids but has minimal effect on vertebrates. In these studies, aphids' settling preferences, host acceptance, feeding, body size and life history parameters were assessed under a range of conditions [24–28]. A common trend across many studies has emerged that showed varying responses depending on the aphid–grass–endophyte combination. For instance, Siegel, et al. [29] assessed a range of grasses, cultivars and endophytes and showed an increase in mortality for *R. padi* and *S. graminum*, which was associated with the presence of lolines, whilst the presence of peramine increased the mortality of *S. graminum* only. Furthermore Meister, et al. [28] found that *R. padi* life span and fecundity (assessed individually) were significantly reduced when feeding on *Epichloë festucae* var. *lolii* infected perennial ryegrass while *M. dirhodum* showed no response. And finally Popay and Cox [30] found certain endophytes decreased *A. lentisci* population size more than others.

There is an additional factor that can influence aphid life history in response to grass endophyte symbiota, namely the methodology used to assess the insecticidal effect on aphids, which makes comparisons between studies difficult. This is not surprising as methods range from detached leaf bioassays to whole plant bioassays, density-dependent to density-independent bioassays, and different bioassay durations [19,24,28,30]. From a routine analytical perspective, it is important to develop a bioassay that is fast and easy to conduct with reproducible results. To date, there are no standard *in-planta* protocols developed to assess the insecticidal effects of grass–endophyte symbiota on aphids, such as those available for synthetic insecticide testing [31,32].

Thus, our study aimed at evaluating the insecticidal effects of grass–endophyte symbiota on life history parameters of aphids using a single seedling bioassay, as a first step towards developing a standard protocol that is rapid, simple, reproducible and scalable, for high throughput screening of many grass–endophyte symbiota combinations and aphid species. Our study also aimed at characterising, for the first time in Australia, the effects of five grass–endophyte symbiota (four commercial and one standard/wild type) on the life history of two species of aphid, *D. noxia* and *A. lentisci* (foliar and root feeding species respectively), and the timeframe in which these symbiota show the strongest effect.

## Materials and methods

### Aphids

*Diuraphis noxia* and *A. lentisci* were collected in South Eastern Australia from a range of locations and host plants (Table 1). *Diuraphis noxia* were collected in June 2016 from barley *Hordeum vulgare* (L.), while *A. lentisci* were collected in August 2017 from glasshouse populations on perennial ryegrass cv. Impact 04 (Table 1). These species were selected because they both represent potential or established foliar and root pests of perennial ryegrass in Victoria, Australia [30,33,34].

Ten *D. noxia* individuals were isolated from a field plant and reared on separate barley cv. Hindmarsh. One clone was selected, and its progeny maintained on barley and perennial ryegrass cv. Alto (the latter without endophyte). Eight *A. lentisci* individuals were isolated from glasshouse plants and reared on separate perennial ryegrass cv. Alto only (the latter also without endophyte). Colonies were established from one clonal lineage per species, maintained on mature plants of barley and perennial ryegrass (*D. noxia*) and perennial ryegrass (*A. lentisci*). Each aphid colony was reared for 4–6 weeks, at which point twenty individuals from the colony were transferred to a new mature plant in an insect-free cage. These aphids were used for all subsequent life history bioassays. From these colonies, twenty to thirty adult aphids were kept individually on barley and perennial ryegrass (*D. noxia*), sixty on perennial ryegrass (*A. lentisci*) and left to produce young. Each day newborn nymphs (1–2 newborn aphids/aphid/day) were transferred to the cup assays. In total there were up to 60 newborn aphids/day for both species. Of these, 24 (*D. noxia*) and 48 (*A. lentisci*) were used in the life history assays per treatment. These newborn aphids were collected over a period of 1–2 weeks approximately to a total of 24 and 168 *D. noxia* from barley and perennial ryegrass respectively, and 288 *A. lentisci* from perennial ryegrass (this meant that not all the assays started the same day).

To confirm aphid species identification, a molecular-based identification method was carried out using the barcode region of the *cytochrome oxidase subunit 1* (CO1) gene [35]. Aphid DNA was extracted using Bio-Rad Chelex© 100 Resin following the method of Walsh et al. [36], with minor modifications: individual aphids were placed in 1.5ml Eppendorf tubes containing 2 glass beads and 20μl of Proteinase K. The contents of the tubes were crushed in a mixer mill for 1 min at 30hz., then 150μl of 5% Chelex© 100 Resin (BioRad) was added and the extract was incubated at 55˚C for 1hr, then at 85˚C for 8 min. A section of the CO1 gene was amplified using the primers LCO1490 (5′–GGTCAACAAATCATAAAGATATTGG–3′) and HCO2198 (5′–TAAACTTCAGGGTGACCAAAAAATCA–3′) [35]. The PCR was performed in 25 μL reaction volumes containing: 1x bovine serum albumin (NEB), 10x Immobuffer (Bioline), 2.5 mM dNTP (Qiagen), 10 μM of each primer, 5 units/μL of Immolase DNA polymerase (Bioline) and 5 μL of template DNA. The cycling conditions were: 94˚C for 6 min; 40 cycles of 94˚C for 30 sec, 51˚C for 50 sec and 72˚C for 50 sec; followed by 2 min at 72˚C. PCR products were sequenced by Macrogen Inc. (Seoul, Korea) and sequences were compared to public databases (NCBI BLASTn), determining similarity and coverage with previously identified species. Sequences were submitted to GenBank (Table 1), and specimens were preserved in 70% and 100% ethanol and deposited in the Victorian Agricultural Insect Collection and Victoria Agricultural Insect Tissue Collection (VAIC and VAITC) at AgriBio, Bundoora, Australia.

## Plants

Perennial ryegrass cv. Alto seed was sourced from Agriseeds (Christchurch, New Zealand) and barley cv. Hindmarsh seed was sourced from Seednet (Horsham, Australia). Perennial ryegrass WE (without endophyte) and barley seeds were germinated on petri dishes with moistened 90mm filter paper (Whatman™) at room temperature (approximately 23˚C and 63% RH). After seven days, seedlings were transferred to potting mix in pots and grown in a controlled environment room at 20˚±2˚C and 62.0±5% RH, and a photoperiod of 14h light: 10h dark until maturity and placed in an insect-free cage (W24.5 x D24.5 x H63.0 cm; 150 x 150μm mesh size; Bugdorm, MegaView Science, Pty. Ltd. Taiwan). These plants were used for maintaining the aphid colonies.

The same germination method was used to grow seedlings of perennial ryegrass (+/- endophytes) for life history bioassays. Bioassays utilised a cup-based system [37], whereby one-

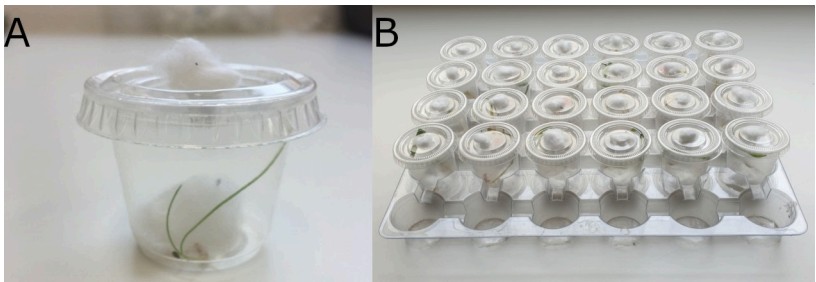

**Fig 1. High throughput *in-planta* bioassay equipment.** (A) An image of an individual bioassay cup containing a seedling with an aphid and moist cotton wool around the roots. (B) Bioassay tray setup containing multiple bioassay cups.

week-old seedlings were placed into individual 30mL plastic cups (Olympus Packaging Pty. Ltd.) with a small piece of moistened cotton wool surrounding the root system. The lid of each cup was perforated to create a hole, and a small piece of cotton wool was placed in each hole to allow evaporation of excess humidity (Fig 1A). Cups were placed in trays that fit up to 30 cups (Fig 1B). The seedlings were watered every few days and were replaced every 7 days during bioassays.

### Endophytes

Perennial ryegrass seeds (see *Plants* section) were infected with fungal endophytes of the species *Epichloë festucae* var. *lolii*. These included four different commercial endophyte strains (AR1, AR37, NEA2 and NEA6) (Table 2) and one wild type or standard endophyte (SE), which is found naturally in many perennial ryegrass pastures [38]. These *Epichloë* endophytes produce a range of alkaloids [Indole Diterpenes (lolitrem B and janthitrem I), Ergopeptines (ergovaline) and Polyketides (peramine)] (Table 2), resulting in each endophyte having a unique alkaloid profile (Table 2).

The presence of endophytes in seed batches and the identity of endophyte strains were confirmed using a Competitive Allele Specific PCR (KASP) bioassay [39]. KASP primers were previously designed and generated by AgriBio and GeneWorks based on allelic variation of single nucleotide polymorphisms (SNP) that differentiate endophytes from one another (SE, AR1, AR37, NEA2 and NEA6). The SNP allele-specific detection is based on a homogeneous fluorescence bioassay, with each forward primer incorporating one distinct fluorescent dye (HEX and FAM) specific for each SNP. The PCR reaction was performed in 10.14 μL reaction volumes containing: 5 μL of 2 x KASP master mix (GeneWorks), 0.14 μl of KASP bioassay mix (GeneWorks) and 5 μl of template DNA. The cycling conditions were: 94˚C for 15 minutes; 9 cycles of 94˚C for 20 seconds and 61˚C for 60 seconds (drop—0.6˚C / cycle); 26 cycles of 94˚C

**Table 2. *Epichloë festucae* var. *lolii* endophyte strains used in this study and associated alkaloid profiles.**

| Alkaloid class | Alkaloid type | SE | AR1 | AR37 | NEA2 | NEA6 |
|---|---|---|---|---|---|---|
| Ergopeptide | Ergovaline | P | * | * | P | P |
| Polyketide | Peramine | P | P | * | P | P |
| Indole Diterpene | Janthitrem I | * | * | P | * | * |
| | Lolitrem B | P | * | * | T | * |

The host plant for all endophytes was perennial ryegrass cv. Alto. SE = standard endophyte; AR1, AR37, NEA2, NEA6 = commercial endophytes. P = alkaloid present
* = alkaloid absent; T = alkaloid present only in trace levels.

20 seconds and 55˚C for 60 seconds; followed by 37˚C for 60 seconds. The data was visualised and analysed using Bio-Rad CFX manager 3.1 software to detect fluorescence and discriminate allelic variation between endophytes.

## Aphid life history parameters

A total of 168 new-born *D. noxia* nymphs and 288 new-born *A. lentisci* nymphs were collected (24 newborn nymphs x 1 aphid species x 5 endophyte treatments and 2 controls; 48 newborn nymphs x 1 aphid species x 5 endophyte treatments and 1 control = 456 newborn nymphs total) and placed individually into new bioassay cups with 7-day-old perennial ryegrass seedlings with and without endophytes (Table 2). *Diuraphis noxia* were placed on to leaves, while *A. lentisci* were placed onto roots. Barley was included as an additional control for *D. noxia* as it is a common host plant for *D. noxia*, where it exhibits high levels of fecundity and survival [40]. This provided assurance of the health of the *D. noxia* clone selected. Life history data of *D. noxia* on barley was not included in statistical analyses. Every 7 days (up to 28 days) new 7-day old seedlings were provided to the aphids. When replacing seedlings, aphids were gently transferred to the new seedlings by lightly brushing the aphid dorsum with a fine paintbrush until they withdrew their stylets. Aphids were then picked up using the same paintbrush and placed near the leaf or the root of the new seedling.

The parameters investigated in this study were mortality and fecundity. In order to calculate these parameters, data was collated on development (whether aphids were nymphs or adults), achieved by recording the number of moults and, time to reproduction (number of days from birth to first reproduction) used to calculate the intrinsic rate of increase ($r_m$). Fecundity was recorded as the number of nymphs born every 24 hours, which were also removed daily, and mortality as the number of aphids that died each day. The intrinsic rate of increase ($r_m$) was calculated using the following formula:

$$r_m = 0.738 \times \{ln(FD)/(D)\}$$

Where FD is the number of nymphs produced during a time equal or greater than the pre-reproductive period and D is the pre-reproductive period (number of days from birth to first reproduction) [41]. Aphids were transferred onto a new seedling, every 7 days. All experiments were carried out in a controlled environment room at 20˚±2˚C and 62.0±5%RH, and a photoperiod of 14h light: 10h dark. All experiments were carried out over a 28 day period from the moment of birth; 28 days was chosen for the duration of the bioassay as it represents a major portion of the average reproductive lifespan of *D. noxia* [40] and *A. lentisci* on perennial ryegrass [30] at 20˚C.

## Statistical analyses

*Diuraphis noxia* nymph mortality was grouped into four time periods of 0–24 (24 hrs), 24–48 (48 hrs), 48–72 (72 hrs) and >72 (>72 hrs) hours, while *A. lentisci* nymph mortality was grouped into four time periods of 0–48 (48 hrs), 48–96 (96 hrs), 96–144 (144 hrs), and >144 (>144 hrs) hours. Adult mortality was grouped into time periods of 14, 21 and 28 days for *D. noxia* and *A. lentisci*. Daily observed fecundity data was grouped into time periods of 14 and 21 days for *D. noxia* and 14, 21 and 28 days for *A. lentisci*. Fecundity and $r_m$ were both assessed twice. First considering all aphids in each treatment and then considering only aphids that survived to reproduction. Data on fecundity and $r_m$ was analysed using a one-way analysis of variance. Differences between treatment means were examined using the least significant difference (LSD) at a 5% level of significance. The unit of analysis was individual aphids in plastic cups (up to 24 and 48 replicates for *D. noxia* and *A. lentisci* respectively). Residuals

versus fitted values plots were examined to determine any need for data transformation to ensure the normality of residuals with constant variance. The difference in mortality of aphid nymphs and adults between treatments at each time period was analysed using logistic regression models, where the number of aphid deaths at each time period was the response variable (success) and logit was the link function. The cumulative mortality of nymphs was also analysed using logistic regression models, where the number of nymphs dead was response and Treatment*Day was the full model, with logit as link function. These models were used to compute the probability of aphid mortality in each time period on each treatment and compare between treatments and time periods. The Wald chi-squared test was used to include/exclude a term in the model. All data in this study was analysed using GenStat version 18 [42].

## Results

### Molecular results

Aphid species identification using the barcode region of the CO1 gene confirmed the identity of *D. noxia* and *A. lentisci* with 100% similarity to previously observed and catalogued DNA sequences.

Endophyte presence in seed batches was confirmed using strain-specific Competitive Allele Specific PCR (KASP) assay, with results showing high incidence in all seed batches. Of approximately 150 seedlings tested for each treatment all were shown to have 100% incidence of the expected endophyte, indicating that endophyte presence is very high in the entire seed batch.

### Effects of *Epichloë festucae* var. *lolii* endophyte symbiota on life history of Diuraphis noxia

*Epichloë festucae* var. *lolii* endophyte symbiota had a significant effect on the mortality of *D. noxia*, most notably at the nymphal stage (Table 3). The average mortality at the nymphal stage was 91% across all endophyte treatments. The total average nymph mortality was highest on aphids tested on SE and NEA2 (both caused 100% nymph mortality) followed by AR1 (96%), NEA6 (88%) and AR37 (71%). The endophyte treatments SE, NEA2 and AR1 resulted

**Table 3. Mortality of *Diuraphis noxia* on all endophyte and endophyte-free treatments observed at the nymphal stage (24, 48, 72 and >72 hrs) and the adult stage (14, 21 and 28 days).**

| Mortality | Barley | WE | SE | AR1 | AR37 | NEA2 | NEA6 | Total mortality [a] | LSD | P-value[b] |
|---|---|---|---|---|---|---|---|---|---|---|
| **Nymph mortality** | 4/21 (0.19) | 8/22 (0.36) | 24/24 (1.00) | 23/24 (0.96) | 17/24 (0.71) | 24/24 (1.00) | 21/24 (0.88) | 0.91 | 0.17 | **<0.001** |
| **Nymph mortality (24 hrs)** | 0/21 (0.00) | 0/22 (0.00) | 4/24 (0.17) | 7/24 (0.29) | 4/24 (0.17) | 9/24 (0.38) | 7/24 (0.29) | 0.26 | 0.22 | **0.004** |
| **Nymph mortality (48 hrs)** | 0/21 (0.00) | 2/22 (0.09) | 7/24 (0.29) | 9/24 (0.38) | 3/24 (0.12) | 10/24 (0.41) | 8/24 (0.33) | 0.31 | 0.24 | **0.029** |
| **Nymph mortality (72 hrs)** | 2/21 (0.10) | 2/22 (0.09) | 6/24 (0.25) | 5/24 (0.21) | 5/24 (0.21) | 2/24 (0.08) | 2/24 (0.08) | 0.17 | 0.2 | 0.337 |
| **Nymph mortality (>72 hrs)** | 2/21 (0.10) | 4/22 (0.18) | 7/24 (0.29) | 2/24 (0.08) | 5/24 (0.21) | 2/24 (0.08) | 4/24 (0.17) | 0.17 | 0.21 | 0.509 |
| **Adult mortality** | 17/21 (0.81) | 16/22 (0.67) | * | 01/24 (0.04) | 7/24 (0.29) | * | 3/24 (0.12) | 0.15 | 0.17 | **<0.001** |
| **Adult mortality (14 days)** | 3/21 (0.14) | 1/22 (0.04) | * | 0/24 (0.00) | 0/24 (0.00) | * | 0/24 (0.00) | 0 | 0.03 | 0.606 |
| **Adult mortality (21 days)** | 7/21 (0.33) | 6/22 (0.25) | * | 0/24 (0.00) | 4/24 (0.17) | * | 3/24 (0.13) | 0.1 | 0.14 | **<0.001** |
| **Adult mortality (28 days)** | 7/21 (0.38) | 9/22 (0.38) | * | 1/24 (0.04) | 3/24 (0.13) | * | 0/24 (0.00) | 0.05 | 0.12 | **<0.001** |

Results are shown as proportions.

[a] Average mortality on endophyte treatments (excluding WE).

[b] p values were calculated using a logistic regression analysis.

* No aphids survived. WE = without endophyte; SE = standard endophyte; AR1, AR37, NEA2, NEA6 = commercial endophytes. Barley data is shown but was not included in the statistical analysis. All bioassays were carried out for 28 days from the moment of birth.

in significantly higher mortality than AR37 ($P < 0.001$). All endophyte treatments resulted in significantly higher nymph mortality than the control (36%).

The total average nymph mortality was highest at 24 and 48 hrs (26% and 31% respectively) and lowest at 72 and >72 hrs (17% in both cases) (Table 3). At 24 and 48 hrs, NEA2, NEA6 and AR1 caused the highest mortality by comparison to all other endophyte treatments and the control WE while at 72 hrs and >72 hrs there was no significant difference between treatments. The rate of mortality (cumulative mortality at 24, 48, 72, >72 hrs) was assessed to determine the time period (early or late nymphal stage) at which treatments had the greatest mortality (Fig 2). Observations of the rate of mortality indicated that gradients of NEA2, NEA6 and AR1 followed a linear-logarithmic gradient (higher mortality at earlier time periods), whereas SE and AR37 followed a linear-exponential gradient (higher mortality at later time periods). Statistically there was no significant difference in the rate or mortality between treatments (Wald statistics $P = 0.269$).

The average mortality at the adult life stage was 15% across all endophyte treatments (Table 3). There was no adult mortality calculated for SE or NEA2 as no aphid survived to the adult life stage on these treatments. At 14 days, there was no significant difference between treatments. At 21 and 28 days, the highest mortality was observed on WE (25% and 38% for each time period respectively) and was significantly different from all other endophyte treatments at 28 days ($P < 0.001$).

The average fecundity was 1.3 nymphs per adult per day across all endophyte treatments, when assessing all aphids in each treatment (Table 4). There was no fecundity calculated for SE or NEA2 as no aphid survived to the adult life stage on these treatments. At 14 and 21 days AR1 exhibited the strongest reduction in fecundity (0.04 and 0.2 nymphs per adult respectively), followed by AR37 (1.5 and 2.8 respectively) and NEA6 (1.4 and 2.0 respectively). All endophyte treatments exhibited significantly reduced fecundity, compared to the control ($P < 0.001$), but there was no significant difference between endophyte treatments. In addition, we carried out a statistical analysis considering the aphids that survived to reproductive age with results showing no significant differences in fecundity between treatments (Table 4). The fecundity rate (cumulative fecundity at 7–21 days) was assessed to determine the time

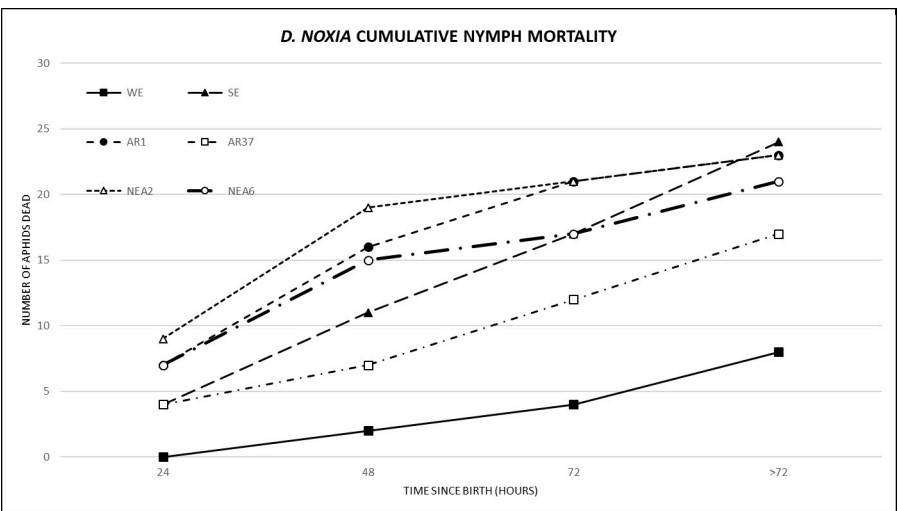

**Fig 2. Cumulative nymph mortality of *Diuraphis noxia* on all endophyte and endophyte-free treatments observed at 24, 48, 72 and >72 hrs.** Results are shown as the cumulative total number of nymphs dead for each time period. WE = without endophyte; SE = standard endophyte; AR1, AR37, NEA2, NEA6 = commercial endophytes.

**Table 4. Fecundity of *Diuraphis noxia* on all endophyte treatments.**

| Fecundity (all aphids) | Barley | WE (n = 22) | SE (n = 24) | AR1 (n = 24) | AR37 (n = 24) | NEA2 (n = 24) | NEA6 (n = 24) | Total fecundity[a] | Standard Error | LSD | P-value[b] |
|---|---|---|---|---|---|---|---|---|---|---|---|
| **Total fecundity** | 23.2 | 7.1 | * | 0.1 | 2.1 | * | 1.7 | 1.3 | n/a | n/a | n/a |
| **Fecundity (14 days)** | 18.1 | 5.1 (±0.66) | * | 0.04 (±0.63) | 1.5 (±0.63) | * | 1.5 (±0.63) | 1.0 | ±0.9 | 1.8 | **<0.001** |
| **Fecundity (21 days)** | 28.3 | 10.0 (±1.11) | * | 0.2 (±1.06) | 2.8 (±1.06) | * | 2.1 (1.06) | 1.7 | ±1.5 | 2.9 | **<0.001** |
| $r_m$ | 0.29 | 0.14 (±0.02) | * | 0.003 (±0.01) | 0.05 (±0.01) | * | 0.03 (±0.01) | 0.03 | ±0.02 | 0.04 | **<0.001** |
| Fecundity (reproducing aphids) | Barley (n = 17) | WE (n = 14) | SE | AR1 (n = 1) | AR37 (n = 7) | NEA2 | NEA6 (n = 3) | Total fecundity[a] | Standard Error | LSD | P-value[b] |
| **Total fecundity** | 28.7 | 11.2 | * | 2.5 | 7.4 | * | 13.7 | 7.8 | n/a | n/a | n/a |
| **Fecundity (14 days)** | 22.4 | 8.1 (±1.25) | * | 1.0 (±4.68) | 5.0 (±1.77) | * | 12.3 (±2.70) | 5.8 | ±3.94 | 8.19 | 0.092 |
| **Fecundity (21 days)** | 35 | 15.6 (±2.00) | * | 4.0 (±7.49) | 9.7 (±2.83) | * | 17.0 (±4.33) | 9.9 | ±6.30 | 13.11 | 0.192 |
| $r_m$ | 0.36 | 0.21 (±0.02) | * | 0.07 (±0.08) | 0.19 (±0.03) | * | 0.25 (±0.05) | 0.17 | ±0.07 | 0.14 | 0.296 |

Results are shown as means.

[a] Average fecundity per female per day on endophyte treatments (excluding WE).

[b] p values were calculated using a one-way analysis of variance.

* No aphids survived. WE = without endophyte; SE = standard endophyte; AR1, AR37, NEA2, NEA6 = commercial endophytes. Barley data is shown but was not included in the statistical analysis.

period at which treatments had the greatest effect on fecundity (Fig 3). Observations of the fecundity rate indicated that AR1 had a low gradient (low no. of nymphs born) over the 7-21-day period, whereas AR37 and WE had a medium gradient (medium no. of nymphs born), and NEA6 had a high gradient (high no. of nymphs born). The fecundity rate was delayed in aphids on AR1 as reproduction began at 13 days, compared to AR37, NEA6 and WE, where

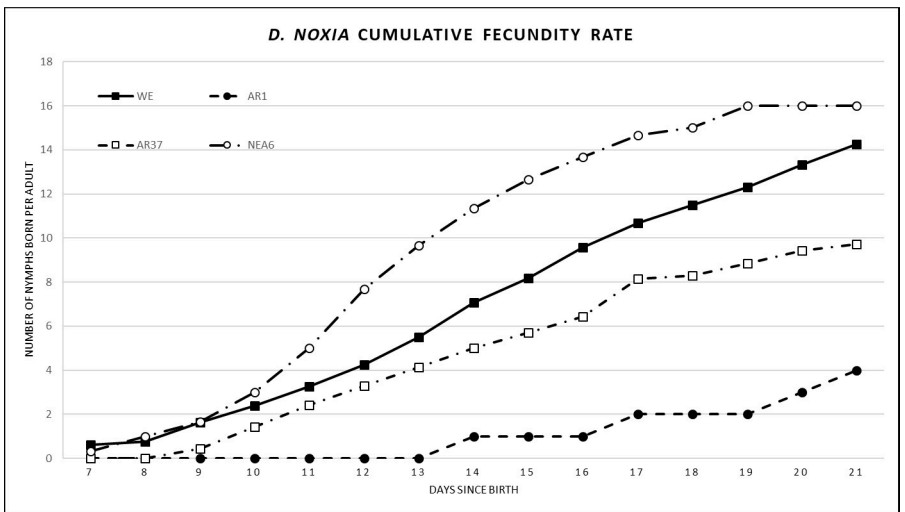

**Fig 3. Cumulative fecundity of *Diuraphis noxia* on all endophyte and endophyte-free treatments observed over 7–21 days.** Results are shown as the average of newborn nymphs per female per day based on aphids that reproduced. Sample sizes are: WE (n = 16), AR1 (n = 1), AR37 (n = 7) and NEA6 (n = 3). Averages of fecundity based on aphids that reproduced is shown in Table 4. WE = without endophyte; AR1, AR37, NEA6 = commercial endophytes.

reproduction began at 7 days. The fecundity rate was increased in aphids on NEA6, particularly between 11–19 days, as the number of nymphs born was consistently above WE.

The average intrinsic rate of increase ($r_m$) was 0.03 across all endophyte treatments, when assessing all aphids in each treatment (Table 4). There was no $r_m$ calculated for SE or NEA2 as no aphid survived to the adult life stage on these treatments. AR1 exhibited the strongest reduction in $r_m$ (0.003), followed by NEA6 (0.03) and AR37 (0.05). All endophyte treatments exhibited significantly reduced $r_m$ compared to the control ($P < 0.001$). There was no significant difference between endophyte treatments. There was no significant difference in $r_m$ between treatments when assessing only the aphids that survived to reproductive age (Table 4).

## Effects of *Epichloë festucae* var. *lolii* endophyte symbiota on life history of *Aploneura lentisci*

*Epichloë festucae var. lolii* endophyte symbiota significantly affected the mortality of *A. lentisci*, and as in *D. noxia*, especially affected the nymphal stage (Table 5). A total of 89% of aphids died during the nymphal stage on all endophyte treatments. The total average nymph mortality was highest on aphids tested on NEA6 and SE (both causing 94% nymph mortality) followed by AR37 and NEA2 (both 88%) and AR1 (83%). All endophyte treatments had significantly higher nymph mortality than the control (69%) ($P = 0.008$), but there was no significant difference between endophyte treatments.

The total average nymph mortality was highest at 96 hrs and 144 hrs (26% and 28% respectively) and lowest at 48 hrs and >144 hrs (17% and 19% respectively) (Table 5). The mortality rates differed over time depending on endophyte treatment. At 48 and 96 hrs NEA6 caused the highest mortality (33% and 40%), although at 48 hrs this was not different from the control. At 144 hrs AR1, caused the highest mortality (50%) while at >144 hrs AR37 showed the highest mortality (29%) of the endophyte group but not of all treatments as the control WE showed the highest mortality (44%). The rate of mortality (cumulative mortality at 48, 96, 144 and >144 hrs) was assessed to determine the time period (early or late nymphal stage) at which treatments had the greatest effect on mortality (Fig 4). Observations of the rate of mortality indicated that gradient of NEA6 followed a linear-logarithmic gradient (higher mortality at earlier time periods), whereas NEA2, SE and AR37 followed a linear gradient (consistent mortality across all time periods), and AR1 and WE followed a linear-exponential gradient (higher mortality at later time periods). Statistically there was no significant difference in the rate or mortality between treatments (Wald statistics $P = 0.060$).

Adult mortality was low in comparison to nymph mortality (11%) (Table 5). At 14 days, mortality was not significantly different between treatments but there were significantly higher mortality rates for WE at 21 and 28 days compared to many endophyte treatments ($P < 0.001$ and $P = 0.007$ respectively).

Average fecundity of *A. lentisci* was significantly reduced to 0.6 nymphs per adult per day across all endophyte treatments, when assessing all aphids (Table 6). For all observation times i.e., 14, 21 and 28 days, endophyte associated mortality was significantly different from the control (except for AR1 at 14 days) ($P < 0.001$) (Table 6). AR37, NEA2 and NEA6 exhibited the strongest reduction in fecundity by comparison to AR1 but also SE showed a strong reduction in fecundity. The fecundity rate (cumulative fecundity at 7–21 days) was assessed to determine the time period at which treatments had the greatest effect on fecundity (Fig 5). Observations of the fecundity rate indicated that NEA2, SE, AR37 and NEA6 all had a low gradient (low no. of nymphs born) over the 7-21-day period, whereas AR1 had a medium gradient (medium no. of nymphs born), and WE had a high gradient (high no. of nymphs born).

**Table 5. Mortality of *Aploneura lentisci* on all endophyte and endophyte-free treatments observed at the nymphal stage (48, 96, 144 and >144 hrs) and the adult stage (14, 21 and 28 days).**

| Mortality | WE | SE | AR1 | AR37 | NEA2 | NEA6 | Total mortality [a] | LSD | P-value[b] |
|---|---|---|---|---|---|---|---|---|---|
| **Nymph mortality** | 33/48 (0.69) | 45/48 (0.94) | 40/48 (0.83) | 42/48 (0.88) | 42/47 (0.88) | 45/48 (0.94) | 0.89 | 0.13 | **0.008** |
| **Nymph mortality (48 hrs)** | 11/48 (0.23) | 6/48 (0.13) | 1/48 (0.02) | 10/48 (0.20) | 8/47 (0.17) | 16/48 (0.33) | 0.17 | 0.15 | **<0.001** |
| **Nymph mortality (96 hrs hrs)** | 0/48 (0.00) | 14/48 (0.29) | 8/48 (0.17) | 9/48 (0.19) | 11/47 (0.23) | 19/48 (0.40) | 0.26 | 0.15 | **<0.001** |
| **Nymph mortality (144 hrs)** | 1/48 (0.02) | 14/48 (0.29) | 24/48 (0.50) | 9/48 (0.19) | 17/47 (0.35) | 3/48 (0.06) | 0.28 | 0.15 | **<0.001** |
| **Nymph mortality (>144 hrs)** | 21/48 (0.44) | 11/48 (0.23) | 7/48 (0.15) | 14/48 (0.29) | 6/47 (0.13) | 7/48 (0.15) | 0.19 | 0.16 | **0.002** |
| **Adult mortality** | 15/48 (0.31) | 3/48 (0.06) | 8/48 (0.17) | 6/48 (0.125) | 6/47 (0.125) | 3/48 (0.06) | 0.11 | 0.13 | **0.008** |
| **Adult mortality (14 days)** | 2/48 (0.04) | 3/48 (0.06) | 0/48 (0.00) | 1/48 (0.02) | 3/47 (0.06) | 0/48 (0.00) | 0.03 | 0.07 | 0.115 |
| **Adult mortality (21 days)** | 8/48 (0.17) | 0/48 (0.00) | 6/48 (0.13) | 1/48 (0.02) | 0/47 (0.00) | 2/48 (0.04) | 0.04 | 0.08 | **<0.001** |
| **Adult mortality (28 days)** | 5/48 (0.10) | 0/48 (0.00) | 0/48 (0.00) | 3/48 (0.06) | 1/47 (0.02) | 0/48 (0.00) | 0.02 | 0.06 | **0.007** |

Results are shown as proportions.

[a] Average mortality on endophyte treatments (excluding WE).

[b] p values were calculated using a logistic regression analysis. WE = without endophyte; SE = standard endophyte; AR1, AR37, NEA2, NEA6 = commercial endophytes. All bioassays were carried out for 28 days from the moment of birth.

The fecundity rate was delayed in aphids on AR37, NEA2 and NEA6 as reproduction began at 9–11 days, compared to WE, SE and AR1, where reproduction began at 7 days. The fecundity rate was slightly increased in aphids on AR1 between 11–14 days, as the number of nymphs born was above that of WE.

The average intrinsic rate of increase ($r_m$) was 0.01 across all endophyte treatments, when assessing all aphids (Table 6). SE and NEA6 exhibited the strongest reduction in $r_m$ (both 0.005), followed by AR37 and NEA2 (both 0.01) and AR1 (0.03). All endophyte treatments exhibited significantly reduced $r_m$ compared to the control, and SE and NEA6 exhibited significantly reduced $r_m$ compared to AR1 ($P < 0.001$). The $r_m$ was 0.11 across all endophyte treatments, when assessing only the aphids that survived to reproductive age (Table 6). NEA6 exhibited the strongest reduction in $r_m$ (0.08), followed by AR37 and NEA2 (0.10), SE (0.11) and AR1 (0.17). Only NEA6 exhibited significantly reduced $r_m$ compared to the control ($P < 0.001$). There was no significant difference between endophyte treatments.

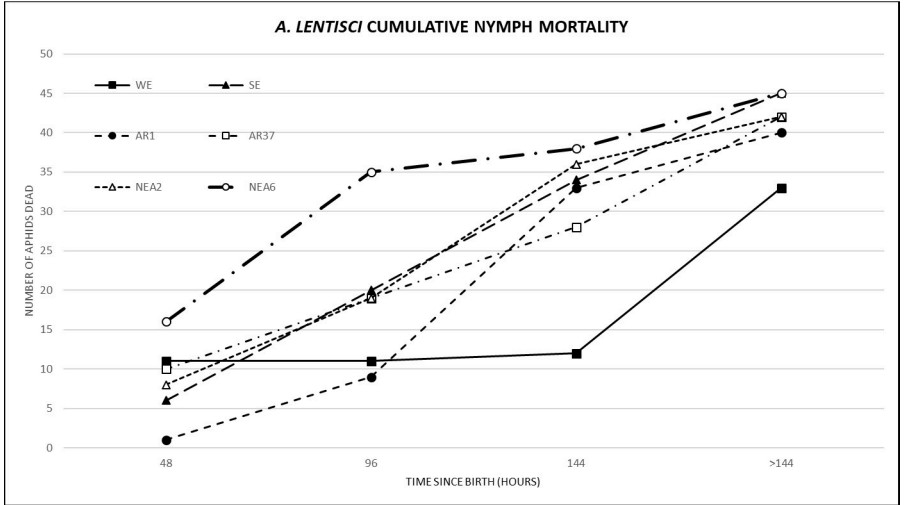

**Fig 4. Cumulative nymph mortality of *Aploneura lentisci* on all endophyte and endophyte-free treatments observed at 48, 96, 144 and >144 hrs.** Results are shown as the cumulative total number of nymphs dead for each time period. WE = without endophyte; SE = standard endophyte; AR1, AR37, NEA2, NEA6 = commercial endophytes.

**Table 6. Fecundity of *Aploneura lentisci* on all endophyte and endophyte-free treatments.**

| Fecundity (all aphids) | WE (n = 48) | SE (n = 48) | AR1 (n = 48) | AR37 (n = 48) | NEA2 (n = 47) | NEA6 (n = 48) | Total fecundity[a] | Standard Error | LSD | P-value[b] |
|---|---|---|---|---|---|---|---|---|---|---|
| Total fecundity | 3.7 | 0.2 | 1.6 | 0.6 | 0.4 | 0.2 | 0.6 | n/a | n/a | n/a |
| Fecundity (14 days) | 1.9 (±0.4) | 0.2 (±0.4) | 1.1 (±0.4) | 0.1 (±0.4) | 0.1 (±0.4) | 0.1 (±0.4) | 0.3 | ±0.4 | 0.8 | **<0.001** |
| Fecundity (21 days) | 4.5 (±0.8) | 0.2 (±0.8) | 1.7 (±0.8) | 0.6 (±0.8) | 0.5 (±0.8) | 0.3 (±0.8) | 0.6 | ±0.8 | 1.58 | **<0.001** |
| Fecundity (28 days) | 4.6 (±0.9) | 0.2 (±0.9) | 1.9 (±0.9) | 1.0 (±0.9) | 0.5 (±0.9) | 0.3 (±0.9) | 0.8 | ±0.9 | 1.8 | **<0.001** |
| $r_m$ | 0.05 (±0.01) | 0.004 (±0.01) | 0.03 (±0.01) | 0.01 (±0.01) | 0.01 (±0.01) | 0.005 (±0.01) | 0.01 | ±0.011 | 0.02 | **<0.001** |
| Fecundity (reproducing aphids) | WE (n = 15) | SE (n = 3) | AR1 (n = 8) | AR37 (n = 6) | NEA2 (n = 6) | NEA6 (n = 3) | Total fecundity[a] | Standard Error | LSD | P-value[b] |
| Total fecundity | 11.7 | 3.3 | 9.4 | 4.7 | 3 | 3.4 | 4.8 | n/a | n/a | n/a |
| Fecundity (14 days) | 6.0 (±0.98) | 3.0 (±2.20) | 6.5 (±1.35) | 1.0 (±1.55) | 1.2 (±1.55) | 1.0 (±2.20) | 2.6 | ±2.4 | 4.8 | **0.014** |
| Fecundity (21 days) | 14.3 (±1.57) | 3.0 (±3.52) | 10.0 (±2.16) | 4.7 (±2.49) | 3.7 (±2.49) | 4.7 (±3.52) | 5.3 | ±3.8 | 7.7 | **0.002** |
| Fecundity (28 days) | 14.9 (±1.89) | 3.0 (±4.23) | 11.6 (±2.59) | 8.3 (±2.99) | 4.2 (±2.99) | 4.7 (±4.23) | 6.4 | ±4.6 | 9.2 | **0.02** |
| $r_m$ | 0.17 (±0.02) | 0.064 (±0.05) | 0.17 (0.03) | 0.08 (±0.03) | 0.06 (±0.03) | 0.08 (±0.05) | 0.11 | ±0.05 | 0.10 | **0.019** |

Results are shown as means.

[a] Average fecundity per female per day on endophyte treatments (excluding WE).

[b] p values were calculated using a one-way analysis of variance. WE = without endophyte; SE = standard endophyte; AR1, AR37, NEA2, NEA6 = commercial endophytes.

## Discussion

### Development of a standardised protocol to determine the insecticidal activity of grass–endophyte symbiota against aphid pests

The bioassay successfully determined the insecticidal activity of grass–endophyte symbiota against *D. noxia* and *A. lentisci*, as results showed aphid life history was significantly negatively

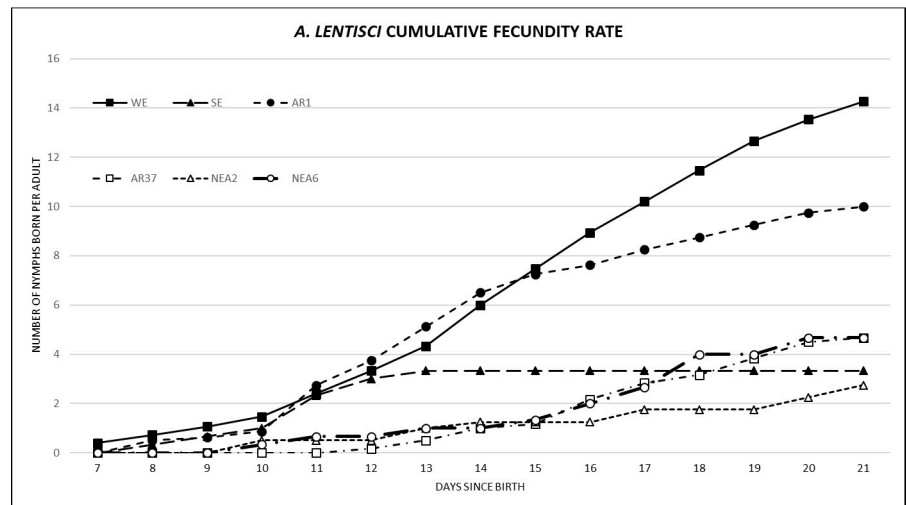

**Fig 5. Cumulative fecundity of *Aploneura lentisci* on all endophyte and endophyte-free treatments observed over 7–21 days.** Results are shown the average of newborn nymphs per female per day based on aphids that reproduced. Sample sizes are: WE (n = 15), SE (n = 3), AR1 (n = 8), AR37 (n = 6), NEA2 (n = 6) and NEA6 (n = 3). Averages of fecundity based on aphids that reproduced is shown in Table 6. WE = without endophyte; AR1, AR37, NEA2, NEA6 = commercial endophytes; SE = standard endophyte.

impacted by all endophyte treatments compared to the control (WE). The bioassay was also able to benchmark endophyte treatments, a key requirement for the establishment of a standardised protocol for comparing grass–endophyte symbiota. The bioassay was also simple, rapid, reproducible, scalable and inexpensive. The simple design consisted of small plastic cups arrayed in a tray, with each cup containing a 7-day-old perennial ryegrass seedling fed on by a single aphid. The assessment procedure was also simple, involving regular aphid life history observations (1 every 24 hours) that were conducted with minimal disturbance by removing the cup lid and visually assessing aphid development, reproduction and mortality. The bioassay was rapid, as a duration of 14 and 21 days was optimal to make meaningful assessments of endophyte insecticidal activities on *D. noxia* and *A. lentisci* respectively. The design was highly reproducible, given there were minimal variables (1 symbiota, 1 aphid) and environmental conditions were kept constant to ensure consistent aphid life history. Contamination by common glasshouse pests that can also influence aphid life history, such as thrips, mites or parasitoids, was avoided through the cup-based design. The compact design of the bioassay was highly scalable, predominantly due to the use of seedlings as opposed to whole plants; this offered the advantage of increasing the number of treatments and replicates while keeping space requirements to a minimum. Finally, the design was also inexpensive, as the materials used were low-cost and readily available. A further advantage of this bioassay is its potential to be deployed as part of the Forage Value Index, which aims to compare perennial ryegrass cultivars in Australia and New Zealand based on yield and heading date, since insecticidal activity is not currently evaluated [43–45]. While the bioassay was developed for perennial ryegrass–*Epichloë* symbiota, it could also potentially be applied to grass–endophyte symbiota across all major pasture species grown in Australia and New Zealand (e.g. Annual ryegrass: *Lolium rigidum–Epichloë occultans*, Tall fescue: *Lolium arundinaceum–Epichloë coenophiala*) [23]. Application of the bioassay could also easily be extended to seed-associated microbes other than *Epichloë* spp., which would offer major benefits considering that biological seed treatments are speculated to capture as much as 20% of the global seed treatment market [46,47]. The bioassay could also be extended to many other species of aphids and possibly other root- and foliar-feeding pasture pests, which are the primary cause of yield loss in pasture systems [48]. Finally, the bioassay could be used to investigate the effects of endophytes on the transmission of plant diseases, given the cup-based assay was originally designed to study aphid-transmitted Luteoviruses [37], as well as to test the effects of other insecticidal seed treatments. Studies are already in progress using this cup-based method and a modified version of it with *A. lentisci* and *R. padi* to investigate the effects of loline-producing alkaloids and entomopathogenic bacterial seed treatments of perennial ryegrass for each species respectively. The versatility of this cup-based assay means it can be used in other areas of research such as life history studies as seen in [49] where the life history of five *R. padi* genotypes was tested on four different host plants and two temperatures.

All *D. noxia* and *A. lentisci* used in this study were sourced from the same clonal populations and therefore lacked genetic variation; however, this bioassay can be applied to multiple aphid populations of different genotypes to determine both inter- and intra-specific differences of endophyte effects on life history [17,50]. In contrast to the clonal aphid populations, the perennial ryegrass cultivars consisted of a mixture of genotypes [1], thus adding an extraneous variable to the study. Previous studies have shown that plant genotype is one of the main factors contributing to both endophyte effectiveness and the alkaloid concentration in a plant [18,20,51–54]. It is therefore possible that individual aphids were exposed to different concentrations of alkaloids throughout their life history, as the seedlings that aphids were reared on were replaced every 7 days. However, the high-replicate design of this bioassay (24–48 replicates) mitigated the potential effect of alkaloid variability caused by plant genotype.

Alkaloid presence was validated from the outset of the study using a strain-specific diagnostic KASP assay, which indicated high incidences of endophytes in seed batches. As such, every seedling evaluated in the bioassay was highly likely to contain an endophyte producing the desired alkaloids. It is also worth noting that although this study involved replacing seedlings every 7 days, observations of seedling health indicated that seedlings in this bioassay design could have been used for 14 days, which would further reduce variability in the alkaloid levels that individual aphids are exposed to throughout their life history. Results also suggest that an experimental period of 14 days, rather than 28 days, would be sufficient to gather significant data on aphid mortality, thus completely eliminating alkaloid variability associated with replacing seedlings.

One final point of note of this study design is its varying suitability to different aphid species. *Diuraphis noxia* reared on the WE control exhibited natural mortality rates, with higher mortality at the adult stage (67%) than at the nymphal stage (36%). This indicated that the design–including the plant species and cultivar–was particularly suited to this aphid species. By contrast, *A. lentisci* reared on the WE control experienced higher mortality at the nymphal stage (69%) than at the adult stage (31%), indicating that the population's life cycle was negatively impacted by a variable or variables other than endophyte treatment, such as the design, plant or cultivar. It is possible that the cup-based design left *A. lentisci* individuals too exposed compared to their natural subterranean habitat, leading to decreased survival at the nymphal stage [30]. Additionally, while every effort was made to transfer aphids gently, the transferral process may still have had a negative effect on aphid health, so eliminating this process may be advantageous. This needs to be tested further. Current research is ongoing into adapting this design to better suit *A. lentisci* life history assays.

## Insecticidal activities of endophytes: Mortality

Mortality was highest in the nymphal stage (i.e. <7 days) for both species, with very few nymphs reaching the adult stage on all endophyte treatments. The nymphal stage has been reported to be more susceptible to endophytes than the adult stage in other aphid species, such as *R. padi* [24]. This finding is of economic importance as early-acting insecticides that are most effective against nymphs will more successfully reduce aphid population growth, due to fewer aphids surviving to reproductive age. The endophyte that showed the strongest effect on nymph mortality in both aphid species was SE, followed by NEA2 (*D. noxia*) and NEA6 (*A. lentisci*). SE produces ergovaline and peramine, as well as high levels of lolitrem B [55,56]. The point of difference between these endophytes is their lolitrem B production: SE produces high levels of lolitrem B, whereas NEA2 only produces trace amounts and NEA6 produce none [17]. The high mortality in *D. noxia* observed on both SE and NEA2 (100%) indicated that aphid mortality occurred regardless of whether lolitrem B was present in high or trace amounts. Lolitrem B is an indole diterpene alkaloid that is most commonly associated with the condition known as ryegrass staggers that affects cattle and sheep when present in high concentrations [57]. The insecticidal mode of action of lolitrem B is not well understood; it is proposed to have either an antibiotic or antixenotic effect [58]. There is evidence of other indole diterpenes (e.g. nodulisporic acids) having insecticidal effects on Diptera [59] and Coleoptera [60], but to date their effect on aphids is unknown. Given that NEA2 produces only trace amounts of lolitrem B and is therefore considered safe for animal consumption, this endophyte may prove an important insecticide against *D. noxia*. For *A. lentisci*, the most insecticidally active endophytes were SE and NEA6, which both produce peramine and ergovaline [61]. Peramine is known to have feeding deterrent effects on certain insects, most notably the Argentine stem weevil, *Listronotus bonariensis* (Kuschel), but these are less potent against aphids

[62]. Conversely, the effect on ergovaline on aphid populations or life history is not well known [17,29,63]. Peramine has been found to have little to no effect on *A. lentisci* from studies with the endophyte AR1 (peramine only) [30]. As such, the insecticidal effect of SE and NEA6 is likely to be derived from ergovaline. Ergovaline is an ergopeptide that is associated with Fescue Foot, a vasoconstrictive condition in grazing livestock that leads to lameness and limb necrosis when ergovaline is present in high concentrations [64]. While the insecticidal activity of ergovaline is known, the mode of action of ergovaline is not well understood [51]. Commercially, NEA6 is regularly sold in conjunction with NEA2 to ensure the pasture contains the insecticidal effects of ergovaline and lolitrem B, but at diluted concentrations to limit any effect on livestock [61]. SE may also act as a viable commercial endophyte under high aphid pressure or when combined with an endophyte that produces no lolitrem B.

## Insecticidal activities of endophytes: Fecundity

Fecundity was reduced by all endophyte treatments. The endophytes that caused the greatest reduction in *A. lentisci* fecundity were SE, NEA2 and NEA6, whereas the endophyte that caused the greatest reduction in *D. noxia* fecundity was AR1. In *D. noxia*, no fecundity was observed on SE and NEA2 as no aphids survived to reproductive age. The common alkaloids associated with these endophyte treatments are lolitrem B, ergovaline and peramine. Other studies have also observed the negative effect of endophytes on aphid fecundity, predominantly endophytes producing lolitrem B and peramine [40, 46]: Meister et al. [28] saw a significant decrease in fecundity of *Rhopalosiphum padi* on perennial ryegrass infected with peramine and lolitrem B-producing endophytes.

Given the high mortality and the low numbers of aphids that reached reproductive maturity, it is likely that the effect of endophytes on fecundity was due to adults experiencing lower general health. The adult and nymph embryo may suffer from malnutrition caused by antixenosis, or the nymph may experience antibiotic effects during gestation within the adult. Our data has already shown that endophytes had a strong effect on nymph mortality; however, it is also worth noting that stillborn aphids were observed on occasion throughout the bioassay–a likely indicator of antibiotic effects during gestation. This is supported by Clement et al. [52], who found *D. noxia* population densities were significantly decreased on four accessions of wild barley infected with endophytic fungi, which they determined was the result of either antibiosis or starvation due to antixenosis. Similarly, when studying *A. lentisci*, Popay & Thom [65] and Popay & Hume [66] found that aphid infestation and population size were consistently lowest on perennial ryegrass infected with certain strains of *Epichloë*. However, in both of these studies it is unclear whether negative effects on adults or nymphs were responsible for these changes in population.

## Translation to field performance

Results of this study indicated that endophyte-free perennial ryegrass is susceptible to infestation by *D. noxia* and *A. lentisci*, highlighting the importance of utilising endophytes in pasture systems under aphid pressure. This study also demonstrated that endophytes have a strong but varying effect on the survival and fecundity of aphids, and this effect is dependent on both endophyte strain and aphid species–a conclusion similar to that of Clement et al. [26,52,67] and Meister et al. [28]. This suggests that a tailored approach to endophyte selection would be required to achieve targeted aphid control, whereas the use of multiple endophytes producing a wide range of alkaloid profiles would be required for effective broad-spectrum control of multiple aphid species. Of the endophytes tested, SE was consistently the most effective at controlling both *A. lentisci* and *D. noxia*, however, it is unsafe for animal consumption and

therefore has limited options for use in pasture production. Nevertheless, it could have commercial potential provided that strategies are used to mitigate lolitrem B toxicity in grazing livestock; for example, through the use of mycotoxin binding agents [68], or by combining SE with other endophytes that produce no lolitrem B (e.g. NEA6). SE is the dominant endophyte profile in unmanaged pasture systems, which suggests it has a clear ecological advantage over other endophyte types. If the risk of ryegrass staggers due to lolitrem B toxicity is too great, a combination of NEA2 and NEA6 would provide adequate broad spectrum control of aphids in pasture systems.

## Conclusions

This study has demonstrated that both *D. noxia* and *A. lentisci* are susceptible to endophytes producing the alkaloids lolitrem B, ergovaline and peramine, and these endophytes have the potential to be used in sustainable pasture management systems to increase production and decrease the use of insecticide sprays. The bioassay we designed can be used as a simple, rapid, reproducible, scalable and inexpensive method of assessing the insecticidal activity of novel endophyte symbiota, and can therefore be used to benchmark commercial and pre-commercial symbiota for use in perennial ryegrass pasture systems. While it may not, in its current form, be entirely suitable for assessment of root-feeding aphids, it is suitable for use with foliar-feeding species and as such, it may significantly accelerate the developmental pipeline from endophyte discovery to endophyte utilisation in dairy farming systems.

## Acknowledgments

We would like to thank and acknowledge the following people and institutions: Sara Nour and Darren Callaway (Agriculture Victoria) for locating *A. lentisci* specimens; Piotr Trebicki (Agriculture Victoria–Grains Innovation Park) for locating and supplying *D. noxia* specimens; Alexander Piper (Agriculture Victoria) for assistance with acquiring GenBank Accession numbers and assistance in aphid colony maintenance; Daniel Lai and Mijail Karpyn (Agriculture Victoria) for assistance in aphid colony maintenance; La Trobe University and Agriculture Victoria for providing equipment and facilities for this project. This work was carried out by N.C. through a postgraduate program.

## Author Contributions

**Conceptualization:** Ross Cameron Mann, German Spangenberg.

**Data curation:** Nicholas Paul Collinson, Khageswor Giri.

**Formal analysis:** Nicholas Paul Collinson, Khageswor Giri.

**Funding acquisition:** Ross Cameron Mann, German Spangenberg.

**Investigation:** Nicholas Paul Collinson, Jatinder Kaur.

**Methodology:** Isabel Valenzuela.

**Project administration:** Ross Cameron Mann.

**Resources:** Ross Cameron Mann.

**Software:** Khageswor Giri.

**Supervision:** Ross Cameron Mann, Isabel Valenzuela.

**Validation:** Ross Cameron Mann, Isabel Valenzuela.

**Visualization:** Nicholas Paul Collinson.

**Writing – original draft:** Nicholas Paul Collinson, Ross Cameron Mann, Isabel Valenzuela.

**Writing – review & editing:** Nicholas Paul Collinson, Ross Cameron Mann, Khageswor Giri, Mallik Malipatil, Jatinder Kaur, German Spangenberg, Isabel Valenzuela.

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
