## [Decision Letter · Decision Letter 0]

1 Nov 2019

PONE-D-19-25662

Novel bioassay to assess antibiotic effects of fungal endophytes on aphids

PLOS ONE

Dear Mr Collinson,

Thank you for submitting your manuscript to PLOS ONE. After careful consideration, we feel that it has merit but does not fully meet PLOS ONE’s publication criteria as it currently stands. Therefore, we invite you to submit a revised version of the manuscript that addresses the points raised during the review process.

Both reviewers were encouraging about this work. However, they noted some elements that needed revision. In particular, I encourage you to revise the methods to meet those details that were noted as missing or unclear. I would also like to see the molecular results that both reviewers highlighted as missing. We look forward to seeing your revision. 

We would appreciate receiving your revised manuscript by Dec 16 2019 11:59PM. To enhance the reproducibility of your results, we recommend that if applicable you deposit your laboratory protocols in protocols.io, where a protocol can be assigned its own identifier (DOI) such that it can be cited independently in the future. For instructions see: http://journals.plos.org/plosone/s/submission-guidelines#loc-laboratory-protocols

We look forward to receiving your revised manuscript.

Kind regards,

Sean Michael Prager, Ph.D.

Academic Editor

PLOS ONE

Journal Requirements:

Reviewers' comments:

Reviewer's Responses to Questions

**Comments to the Author**

1. Is the manuscript technically sound, and do the data support the conclusions?

Reviewer #1: Partly

Reviewer #2: Yes

2. Has the statistical analysis been performed appropriately and rigorously? 

Reviewer #1: N/A

Reviewer #2: Yes

3. Have the authors made all data underlying the findings in their manuscript fully available?

Reviewer #1: Yes

Reviewer #2: Yes

4. Is the manuscript presented in an intelligible fashion and written in standard English?

Reviewer #1: Yes

Reviewer #2: Yes

5. Review Comments to the Author

Reviewer #1: Research is interesting. l re-read this paper numerous times looking for replication of the research and was unable to find it. Thus, I cannot appropriately comment on the science. There are numerous questions I have within the methodology sections (actual set up, sample size, infection methods) as well as the background and reasoning for choosing these insects. I feel that in order for this to be published there should be a replication of the life history events of these aphids like other life history related papers. Once replicated, I do believe this could only require minor revisions in elaborating in certain parts of this writing. Until then, my comments on that part are unnecessary.

Reviewer #2: The authours describe cup-based system suitable for bioassays involving two species of aphids with different feeding locations, seedlings and endophytes. The manuscript is well written, thanks, and the results are justified by the data. Several of the commercially available endophytes seem promising against aphids and I’m always in favour of biological pest control options when they are available. I am not requesting any major revisions but I’ve detailed suggestions, that could be considered minor revisions in the text below.

Keywords. I think the authours have the paper well covered with their keywords. Does PLOSOne have a keyword limit?

Abstract: The abstract is clear and mirrors the findings of the paper.

Introduction: The introduction clearly states the purpose and gives enough background information to make the objectives understandable.

Materials and Methods:

The creation of aphid clones (lines 119-214) was not very clear as written. For example, Line 194 suggests that all aphids used were raised on perennial rye grass…why mention the colony that was started on barely then? Was it your D. noxia aphid lines from barley that were then used as a control on barley in the endophyte experiments or did they come from the ryegrass colony? Lines 150-153 don’t really make it much clearer.

The materials and methods are also unclear at lines 194-200. How many adult aphids were went into plastic cups total? Then at 196, 24 and 48 nymphs were produced…were these per adult total or was it the grand total of nymphs produced? The bracketed set up of nymphs into treatments doesn’t make sense as is written if only 24 and 48 nymphs were produced total. Please make it clearer how you generated the 456 nymphs for the experiments is what I suppose I’m trying to say.

Lines 180-191. The molecular check of the seedlings for the presence of endophytes is not reported on in the results section or mentioned in the discussion. Could you please include a section in the results that details the molecular results please and then would it worthwhile to mention something about the ability of the seedlings in your novel bioassay to pick up the endophytes from the seed treatments and have them consistently expressed (ie. A percentage of infected plants in each treatment, to confirm that the seedlings had endophytes and that the bioassay allows for endophyte infiltration of seedlings?) If you are detecting fluorescence is there an option to report on the time to positive fluorescence as a semi-quantitative method of assessing the titer of endophytes in seedlings?

You’ve mentioned that seedlings were replaced every 7 days but have not described the methods of transferring aphids from one cup to another, or from one seedling to another. Could the handling of aphids have led to increased mortality? Ie. High mortality in nymphs of A. lentici WE control treatment.

Statistics: Fine.

Results:

Add molecular results please (as above)

Tables and Figures are all well done and necessary for data presentation. I liked the description of the shapes of the death curves in Figs 2 and 4.

In Fecundity tables where the mean number of offspring are shown, it would help to add in the standard error of the mean to show how much variability there was among aphids in each treatment. For example (Table 6 WE Total fecundity of 3.7 (±S.E.)

Line 321: That is high mortality of your nymphs on the Without Endophyte (WE) plants. Fig 4 helps to ease my concern because it seems most of them lived over 144 hours but that is a lot of aphids that did not make it to maturity and adulthood to reproduce. In the discussion at 446 – 451 you’ve given some good suggestions on why this bioassay might not be suitable for this species of root feeding aphid. It’s too bad that it wasn’t very suitable because there would be a great benefit to studying the life history of root feeding aphids outside of the soil. So, in light on the high mortality and potential unsuitability of the assay for root-feeding aphids, could you please add a caution to your abstract and conclusion to that affect. Readers might not pick up on the implications of the high nymphal mortality of A. lentisci in the WE treatment if they only read the abstract and skim the manuscript.

Discussion:

It might be useful to detail other aphid studies that might benefit from adoption of your cup bioassay to round off your discussion. For example, other insecticidal seed treatments perhaps, life history studies on other aphid species etc?

6. PLOS authors have the option to publish the peer review history of their article (what does this mean?). If published, this will include your full peer review and any attached files.

Reviewer #1: No

Reviewer #2: No

---

## [Author Response · Author response to Decision Letter 0]

6 Dec 2019

Reviewer #1 comments:

Research is interesting. l re-read this paper numerous times looking for replication of the research and was unable to find it. Thus, I cannot appropriately comment on the science. There are numerous questions I have within the methodology sections (actual set up, sample size, infection methods) as well as the background and reasoning for choosing these insects. I feel that in order for this to be published there should be a replication of the life history events of these aphids like other life history related papers. Once replicated, I do believe this could only require minor revisions in elaborating in certain parts of this writing. Until then, my comments on that part are unnecessary.

Response:

We are unsure if the reviewer is referring to technical replication or biological replication? We assume the reviewer is referring to technical replication i.e., repeating the same assay more than once to test the variability of the protocol itself. In our case this was not feasible due to time constraints. This cup-based bioassay was considered robust enough for obtaining consistent data given results from previous aphid studies. We chose high biological replication (24 and 48 aphids/treatment) to capture the inherent variability of bioassays. 

Reviewer #2 comments:

The authors describe cup-based system suitable for bioassays involving two species of aphids with different feeding locations, seedlings and endophytes. The manuscript is well written, thanks, and the results are justified by the data. Several of the commercially available endophytes seem promising against aphids and I’m always in favour of biological pest control options when they are available. I am not requesting any major revisions, but I’ve detailed suggestions, that could be considered minor revisions in the text below.

Keywords. I think the authors have the paper well covered with their keywords. Does PLOSOne have a keyword limit?

Abstract: The abstract is clear and mirrors the findings of the paper.

Introduction: The introduction clearly states the purpose and gives enough background information to make the objectives understandable.

Materials and Methods:

The creation of aphid clones (lines 119-214) was not very clear as written. For example, Line 194 suggests that all aphids used were raised on perennial rye grass…why mention the colony that was started on barely then? Was it your D. noxia aphid lines from barley that were then used as a control on barley in the endophyte experiments or did they come from the ryegrass colony? Lines 150-153 don’t really make it much clearer.

Response:

We have now re-worded this section (in lines 122-130) to clarify that aphids from the field were placed on barley (D. noxia) or perennial ryegrass (A. lentisci) and that one selected clone was reared on barley and perennial ryegrass (D. noxia) and perennial ryegrass (A. lentisci). Sections previously in lines 162-166 have been moved to this section.

“Ten D. noxia individuals were isolated from a field plant and reared on separate barley cv. Hindmarsh. One clone was selected, and its progeny maintained on barley and perennial ryegrass cv. Alto (the latter without endophyte). Eight A. lentisci individuals were isolated from glasshouse plants and reared on separate perennial ryegrass cv. Alto only (the latter also without endophyte). Colonies were established from one clonal lineage per species, maintained on mature plants of barley and perennial ryegrass (D. noxia) and perennial ryegrass (A. lentisci). Each aphid colony was reared for 4-6 weeks, at which point twenty individuals from the colony were transferred to a new mature plant in an insect-free cage. These aphids were used for all subsequent life history bioassays.”

Reviewer #2 comments:

The materials and methods are also unclear at lines 194-200. How many adult aphids were went into plastic cups total? Then at 196, 24 and 48 nymphs were produced…were these per adult total or was it the grand total of nymphs produced? The bracketed set up of nymphs into treatments doesn’t make sense as is written if only 24 and 48 nymphs were produced total. Please make it clearer how you generated the 456 nymphs for the experiments is what I suppose I’m trying to say.

Response:

We have now re-worded this section (in lines 130-137) to clarify how many adult aphids reproduced and how many 1st instar nymphs were produced in total and per aphid/day and used in the bioassays.

“From these colonies, twenty to thirty adult aphids were kept individually on barley and perennial ryegrass (D. noxia), sixty on perennial ryegrass (A. lentisci) and left to produce young. Each day newborn nymphs (1-2 newborn aphids/aphid/day) were transferred to the cup assays. In total there were up to 60 newborn aphids/day for both species. Of these, 24 (D. noxia) and 48 (A. lentisci) were used in the life history assays per treatment. These newborn aphids were collected over a period of 1-2 weeks approximately to a total of 24 and 168 D. noxia from barley and perennial ryegrass respectively and 288 A. lentisci from perennial ryegrass (this meant that not all the assays started the same day).”

Reviewer #2 comments:

Lines 180-191. The molecular check of the seedlings for the presence of endophytes is not reported on in the results section or mentioned in the discussion. Could you please include a section in the results that details the molecular results please and then would it worthwhile to mention something about the ability of the seedlings in your novel bioassay to pick up the endophytes from the seed treatments and have them consistently expressed (ie. A percentage of infected plants in each treatment, to confirm that the seedlings had endophytes and that the bioassay allows for endophyte infiltration of seedlings?) If you are detecting fluorescence is there an option to report on the time to positive fluorescence as a semi-quantitative method of assessing the titer of endophytes in seedlings?

Response

We have added a new section (in lines 263-268) to clarify the results from the molecular tests carried out in our study i.e., the aphid identification and endophyte presence molecular tests. 

“Aphid species identification using the barcode region of the CO1 gene confirmed the identity of D. noxia and A. lentisci with 100% similarity to previously observed and catalogued DNA sequences.”

Endophyte presence in seed batches was confirmed using strain-specific Competitive Allele Specific PCR (KASP) assay, with results showing high incidence in all seed batches. Of approximately 150 seedlings tested for each treatment all were shown to have 100% incidence of the expected endophyte, indicating that endophyte presence is very high in the entire seed batch.” 

Reviewer #2 comments:

You’ve mentioned that seedlings were replaced every 7 days but have not described the methods of transferring aphids from one cup to another, or from one seedling to another. Could the handling of aphids have led to increased mortality? Ie. High mortality in nymphs of A. lentici WE control treatment.

Response

We have now re-worded this section (in lines 219-222) to clarify how the transfer of aphids onto new seedlings was carried out. We do not think that the handling caused the high mortality of A. lentisci observed in WE, however this needs to be tested further. Perhaps the perennial ryegrass cultivar was not suitable. This is something we are interested in testing, along with others factors such as stylet morphology of the 1st instars.

“Every 7 days (up to 28 days) new 7-day old seedlings were provided to the aphids. When replacing seedlings, aphids were gently transferred to the new seedlings by lightly brushing the aphid dorsum with a fine paintbrush until they withdrew their stylets. Aphids were then picked up using the same paintbrush and placed near the leaf or the root of the new seedling.”

Reviewer #2 comments:

Statistics: Fine. 

Results:

Add molecular results please (as above)

 Response:

Done (in lines 263-268)

Reviewer #2 comments:

Tables and Figures are all well done and necessary for data presentation. I liked the description of the shapes of the death curves in Figs 2 and 4.

In Fecundity tables where the mean number of offspring are shown, it would help to add in the standard error of the mean to show how much variability there was among aphids in each treatment. For example (Table 6 WE Total fecundity of 3.7 (±S.E.)

Response:

We have added the standard errors to Table 4 and Table 6.

Reviewer #2 comments:

Line 321: That is high mortality of your nymphs on the Without Endophyte (WE) plants. Fig 4 helps to ease my concern because it seems most of them lived over 144 hours but that is a lot of aphids that did not make it to maturity and adulthood to reproduce. In the discussion at 446 – 451 you’ve given some good suggestions on why this bioassay might not be suitable for this species of root feeding aphid. It’s too bad that it wasn’t very suitable because there would be a great benefit to studying the life history of root feeding aphids outside of the soil. So, in light on the high mortality and potential unsuitability of the assay for root-feeding aphids, could you please add a caution to your abstract and conclusion to that affect. Readers might not pick up on the implications of the high nymphal mortality of A. lentisci in the WE treatment if they only read the abstract and skim the manuscript.

Response:

We have added cautions to the abstract (lines 27-28), the discussion (lines 485-487) and the conclusions (lines 574-575) mentioning that the assay, in its current form, may not be suitable for A. lentisci or to root-dwelling insects in general, however new methods are currently being developed based on this design i.e., compact with a high throughput capability that will hopefully address this issue for future studies. 

…”however we would caution that it may not be suitable for the assessment of root-feeding aphids, as this species exhibited relatively high mortality on the control as well.”

“Additionally, while every effort was made to transfer aphids gently, the transferral process may still have had a negative effect on aphid health, so eliminating this process may be advantageous. This needs to be tested further.”

“While it may not, in its current form, be entirely suitable for assessment of root-feeding aphids, it is suitable for use with foliar-feeding species and…”

Reviewer #2 comments:

Discussion:

It might be useful to detail other aphid studies that might benefit from adoption of your cup bioassay to round off your discussion. For example, other insecticidal seed treatments perhaps, life history studies on other aphid species etc?

Response:

We proposed some potential studies that could be done using these methods in lines 440-450, but we have now added more detail about specific studies that could benefit from using the cup-bioassay design, and two that are currently in the early stages of implementation that include a study to improve the design for use with root aphids. We have added in lines 450-456 a sentence to address this.

“as well as to test the effects of other insecticidal seed treatments. Studies are already in progress using this cup-based method and a modified version of it with A. lentisci and R. padi to investigate the effects of loline-producing alkaloids and entomopathogenic bacterial seed treatments of perennial ryegrass for each species respectively. The versatility of this cup-based assay means it can be used in other areas of research such as life history studies as seen in (Valenzuela et. al., 2010) where the life history of five R. padi genotypes was tested on four different host plants and two temperatures.”

---

## [Decision Letter · Decision Letter 1]

24 Jan 2020

Novel bioassay to assess antibiotic effects of fungal endophytes on aphids

PONE-D-19-25662R1

Dear Dr. Collinson,

We are pleased to inform you that your manuscript has been judged scientifically suitable for publication and will be formally accepted for publication once it complies with all outstanding technical requirements.

With kind regards,

Sean Michael Prager, Ph.D.

Academic Editor

PLOS ONE

Additional Editor Comments (optional):

Reviewers' comments:

Reviewer's Responses to Questions

**Comments to the Author**

1. If the authors have adequately addressed your comments raised in a previous round of review and you feel that this manuscript is now acceptable for publication, you may indicate that here to bypass the “Comments to the Author” section, enter your conflict of interest statement in the “Confidential to Editor” section, and submit your "Accept" recommendation.

Reviewer #1: All comments have been addressed

Reviewer #2: All comments have been addressed

2. Is the manuscript technically sound, and do the data support the conclusions?

Reviewer #1: Yes

Reviewer #2: Yes

3. Has the statistical analysis been performed appropriately and rigorously? 

Reviewer #1: Yes

Reviewer #2: Yes

4. Have the authors made all data underlying the findings in their manuscript fully available?

Reviewer #1: Yes

Reviewer #2: Yes

5. Is the manuscript presented in an intelligible fashion and written in standard English?

Reviewer #1: Yes

Reviewer #2: Yes

6. Review Comments to the Author

Reviewer #1: I apologize about the minimal comments submitted last time around. Not all of my comments copied and pasted into the boxes- that is entirely my fault and thus there is not much I can criticize here. Thank you for the replication clarification. I do believe moving forward for future experiments you should do more biological reps to account for variabilities in colony life history.

With that, reviewer 2 addressed most of my concerns that I had with the paper. I think the revisions that have been made are sufficient.

I am not sure if this is just a formatting thing with track changes but the spacing on table 4 is odd. The discussion is more robust now with the additions.

Reviewer #2: Thanks for addressing my comments so well. I'm happy with the changes that have been made.

References

It looks like you've made the references with citation manager software so I didn't take a close look at them.

7. PLOS authors have the option to publish the peer review history of their article (what does this mean?). If published, this will include your full peer review and any attached files.

Reviewer #1: No

Reviewer #2: No

---

## [Editor Report · Acceptance letter]

28 Jan 2020

PONE-D-19-25662R1 

Novel bioassay to assess antibiotic effects of fungal endophytes on aphids 

Dear Dr. Collinson:

I am pleased to inform you that your manuscript has been deemed suitable for publication in PLOS ONE. Congratulations! Your manuscript is now with our production department. 

With kind regards,

on behalf of

Dr. Sean Michael Prager 

Academic Editor

PLOS ONE